# Preparation of Solid Lipid Nanoparticle-Ferrous Sulfate by Double Emulsion Method Based on Fat Rich in Monolaurin and Stearic Acid

**DOI:** 10.3390/nano12173054

**Published:** 2022-09-02

**Authors:** Edy Subroto, Robi Andoyo, Rossi Indiarto, Endah Wulandari, Elgi Fadhilah Nadhirah Wadhiah

**Affiliations:** Department of Food Industrial Technology, Faculty of Agro-Industrial Technology, Universitas Padjadjaran, Bandung 45363, Indonesia

**Keywords:** solid lipid nanoparticle, double emulsion, iron, ferrous sulfate, monolaurin

## Abstract

Ferrous sulfate is one type of iron that is commonly used in iron supplementation and fortification in food products, but it has low stability and an unfavorable flavor, causing its use to be limited. Encapsulation in a solid lipid nanoparticle (SLN) system is one technology that offers stable active compound protection and a good delivery system; however, a solid lipid matrix should be selected which has good health effects, such as glycerol monolaurate or monolaurin. The purpose of this study was to obtain SLN-ferrous sulfate based on stearic acid and fat rich in monolaurin. SLN-Ferrous sulfate was synthesized at various concentrations of monolaurin-rich fat (20%; 30%; 40% *w*/*w* lipid) and various concentrations of ferrous sulfate (5%; 10%; 15% *w*/*w* lipid). The results showed that the use of monolaurin-rich fat 40% *w*/*w* lipid and 15% *w*/*w* ferrous sulfate produced the best characteristics with high entrapment efficiency and loading capacity of 0.06%. The Z-average value of SLN was 292.4 nm with a polydispersity index (PI) of 1.03. SLN-ferrous sulfate showed a spherical morphology, where the Fe trapped in the SLN was evenly dispersed in the lipid matrix to form a nanosphere system. Preparation of SLN-ferrous sulfate by double emulsion method based on stearic acid and fat rich in monolaurin effectively encapsulated ferrous sulfate with high entrapment efficiency and good physicochemical properties.

## 1. Introduction

Supplementation and fortification of iron in food are methods that are widely used to overcome the problem of anemia due to iron deficiency [1]. Iron fortification in food products has been widely carried out, especially in flour-based products such as biscuits, cookies, or flakes. The irons commonly used for the fortification of food products are ferrous sulfate, ferrous fumarate, and ferrous gluconate [2]. Ferrous sulfate is widely used as an iron fortification in food products because it is easy to obtain, has high bioavailability, and has a low price, but it has the disadvantage of having a stronger aroma and taste of iron than ferrous fumarate and ferrous gluconate [3,4]. 

The process or technique that can be conducted to overcome the problem of the sharp aroma and taste of iron in ferrous sulfate is encapsulation. The encapsulation technique that is currently of concern is by means of nanotechnology, where micronutrients are trapped in nanoparticles with particle sizes ranging from 10 to 1000 nm, which can be made from surfactants, lipids, proteins, or carbohydrates [5,6]. Solid lipid-based nanoparticles are suitable for protecting active compounds and good delivery systems, namely in the form of solid lipid nanoparticles (SLN) and nanostructured lipid carriers (NLC) [7,8,9]. 

SLN is easier to produce because it does not require the formation of nanostructures in each particle; thus, SLNs are widely developed. Various formulation techniques for the production of SLN have been developed, including using high-pressure homogenization, solvent emulsification-evaporation, emulsification-sonification, solvent injection, solvent diffusion, micro-emulsion, and double emulsion “water in oil in water” (W/O/W) [10]. Some researchers combine it with other techniques, such as the electrospinning method [11,12] and electrospraying method [13,14], but these methods are more suitable for polar encapsulant materials and require high-tech instrumentation. In addition, electrohydrodynamic techniques such as electrospinning and electrospraying use high energy to prepare nanomaterials, which are among other disadvantages [15,16,17,18]. In encapsulation for the encapsulant material in the form of non-polar lipids, while the active ingredients are polar such as ferrous sulfate, the encapsulation by the formation of the double emulsion (W/O/W) assisted with homogenization and ultrasonication is a suitable encapsulation technology and produces high entrapment efficiency through binding by a double emulsifier [19]. Various lipids have also been used, such as stearic acid, trimyristin, tripalmitin, Glyceryl monostearate, cetyl palmitate, and tristearin [10]; however, the use of a solid lipid matrix in the form of saturated fatty acids such as stearic and palmitic is not good for health; therefore, a solid lipid matrix should be selected which is healthier, such as glycerol monolaurate or commonly known as monolaurin [20,21].

Monolaurin is a monoacylglycerol from lauric acid, which is a medium chain fatty acid (MCFA). MCFA from monolaurin can be directly digested and easily absorbed so that it provides a fast source of energy, and is not stored as body fat [22,23]. Monolaurin is a bioactive lipid because it is antiviral, antibacterial, and can improve the immune system [20,24,25]. Monolaurin can also be used as a non-ionic emulsifier in the food and pharmaceutical industries. The monolaurin structure consists of a non-polar hydrocarbon group and a polar hydroxyl group from glycerol, making monolaurin able to interact with both water and oil [21,26]. 

The amphiphilic properties of monolaurin make monolaurin suitable for use as an emulsifier as well as a solid matrix for the synthesis of SLN-iron by the double emulsion; however, SLN preparation using fat rich in monolaurin as a solid matrix as well as an emulsifier with the method of forming a W/O/W double emulsion is still limited, so the use of fat rich in monolaurin is a novelty in this study for SLN fabrication; this study aimed to fabricate SLN-ferrous sulfate using a mixture of stearic acid and monolaurin-rich fat as a lipid matrix as well as an emulsifier using the double emulsion (W/O/W) method by varying the concentration of monolaurin-rich fat and ferrous sulfate concentration. Several parameters were studied, such as particle size, entrapment efficiency, particle morphology, and microstructure of SLN.

## 2. Materials and Methods

### 2.1. Materials

Ferrous sulfate (FeSO_4_.7H_2_O), coconut stearin, stearic acid, Tween 80, glycerol, tert-butanol, NaOH, citric acid, hexane, and other supporting ingredients with analytical grade specifications.

### 2.2. Preparation of Fat Rich in Monolaurin

The preparation of fat rich in monolaurin was carried out by chemical glycerolysis of coconut stearin in a solvent system [27]. A mixture of coconut stearin and glycerol with a molar ratio of 1:5 was melted using a hot plate at 60 °C, and then tert-butanol was added with a ratio of 1:2 (*w*/*v*) and NaOH (3% of the substrate) as a catalyst. Glycerolysis was carried out in a reflux reactor at 170 °C for 4 hours. The reaction mixture was neutralized by adding 20% citric acid, and then hexane with a ratio of 1:4 *v*/*v* was added to extract acylglycerol. The solvent was evaporated using a rotary evaporator, and then the acylglycerol fraction was transferred to a container for analysis and stored in the refrigerator until used.

### 2.3. Preparation of SLN-Ferrous Sulfate

The preparation of SLN-ferrous sulfate was carried out by a solvent-free method that combines the principle of double emulsion (W/O/W) and melts dispersion techniques [19]. Solid fat 6 g, which was a mixture of stearic acid and fat rich in monolaurin that had been produced in the previous stage, was heated above its melting temperature (60 °C) and stirred with a magnetic stirrer (±200 rpm), then 2 mL of ferrous sulfate solution was added. The mixture was homogenized using a high-speed homogenizer for 1 minute, followed by ultrasonication for 2 minutes at an amplitude of 45% (500 W) to form the first emulsion (W_1_/O). The first emulsion was then added with 60 mL of Tween 80, and the mixture was homogenized again with a high-speed homogenizer (60 °C, 7 minutes, 13,000 rpm) and continued with ultrasonication at an amplitude of 45% (500 W) for 4 minutes to form a second emulsion (W_1_/O/W_2_). The double emulsion was poured into 500 mL of cold water (5–10 °C) under a slow magnetic stirrer (±50 rpm) for 5 minutes to increase the solidification of SLN. The SLN sample was then re-ultrasonicated at an amplitude of 45% (500 W) for 4 minutes. SLN in the form of dispersion was then frozen for ± 24 hours, then lyophilized at −50 °C for 72 hours using a freeze dryer. The formulation for making SLN is shown in Table 1.

### 2.4. Analysis of Particle Size and Polydispersity Index

Particle size and polydispersity index (PI) of SLN were measured using a Particle size analyzer (HORIBA SZ-100). Measurements were conducted on SLN samples in the form of 1 µL dispersion of the sample diluted to a concentration of 1 ppm. The samples were analyzed at a scattering angle of 90, and the polydispersity index was expressed as a representation of the Scattering Light Intensity.

### 2.5. Analysis of Entrapment Efficiency and Loading Capacity

Total iron that was successfully encapsulated was determined using atomic absorption spectroscopy (AAS) (Double Beam Spectrophotometer U-2900/2910; Hitachi) by analyzing the supernatant separated from the SLN [28]. The supernatant was destroyed using nitric acid before testing the Fe content using AAS. The value of entrapment efficiency (EE) can be calculated by Equation (1):EE (%) = (Fe_total_ − Fe_free_)/(Fe_total_) × 100%(1)
where Fe_total_ is the amount of Fe used in the formulation, and Fe_free_ is the amount of free Fe in the supernatant. Loading capacity can also be determined by analyzing the Fe content in SLN that has been lyophilized using a freeze dryer. The value of loading capacity (LC) can be calculated by Equation (2):LC (%) = (Entrapped Fe)/(W of lipid) × 100%(2)
where Entrapped Fe is the amount of Fe in the SLNs, and W of lipid is the weight of the vehicle.

### 2.6. Analysis of Morphology and Particle Structure of SLN

The morphology of SLN particles was analyzed using a scanning electron microscope (SEM) (TM3000 Tabletop Microscope; Hitachi, Tokyo, Japan). The lyophilized SLN samples were placed on double-faced carbon tapes (Ted Pella, Inc., Redding, CA, USA) mounted on aluminum stubs. Micrographs were captured at a voltage of 15 kV and a magnification of ×3000.

The SLN particle structure was analyzed using a Transmission Electron Microscope (TEM HT7700 model; Hitachi, Tokyo, Japan) with an accelerating voltage of 100 kV. For measurement, one drop of diluted SLN dispersion was placed on a copper grating. The samples were stained with 5% (*w*/*v*) uranyl acetate solution to enhance the microscope contrast and then coated with a carbon film to avoid degradation under the electron beam.

## 3. Results and Discussion

### 3.1. Characteristics of Fat Rich in Monolaurin

Fat rich in monolaurin was obtained from the chemical glycerolysis of coconut stearin using NaOH as a catalyst. Glycerolysis of coconut stearin converted triacylglycerol into monoacylglycerol and diacylglycerol. The mono- and diacylglycerol produced was dominated by lauric acid, especially in the form of monolaurin, because the highest fatty acid content in coconut oil was lauric acid, about 47.79%. The resulting Fat Rich in monolaurin could be used as an emulsifier in SLN preparation to maintain the stability and homogeneity of ferrous sulfate coated with solid lipid nanoparticles. The characteristics of fat rich in monolaurin can be seen in Table 2.

Based on Table 2, it is known that no TAG content was detected in fat rich in monolaurin resulting from glycerolysis of coconut stearin, while acylglycerol was dominated by MAG (86.45%) and DAG (13.55%). At the same time, the monolaurin content was about 40.52%. The high MAG and DAG values indicated that the coconut stearin glycerolysis reaction was running well, which succeeded in converting TAG into MAG and DAG.

Fat rich in monolaurin had a high emulsification capacity, which was about 95.26%. The high emulsion capacity was caused by the high MAG and DAG produced during the glycerolysis reaction. MAG and DAG could reduce the surface tension between water and oil. The more MAG and DAG formed, the smaller the surface tension and the more stable the emulsion formed [29,30]. The high value of emulsifying capacity indicated that fat rich in monolaurin could be used as an emulsifier in the preparation of SLN-ferrous sulfate as well as a solid lipid matrix in SLN.

### 3.2. Particle Size, Z-Average, Polydispersity Index, and Entrapment Efficiency of SLN-Ferrous Sulfate

The SLN preparation process was carried out using a solvent-free method that combines the principle of double emulsion (W/O/W) and melt dispersion techniques. The SLN formulation for the treatments of C1, C2, and C3 were formulations with varying concentrations of fat rich in monolaurin used. While C3, C4, and C5 were formulations with varying concentrations of ferrous sulfate used. The characteristics of particle size or Z-average, polydispersity index, and entrapment efficiency of SLN-ferrous sulfate can be seen in Figure 1 and Table 3. 

#### 3.2.1. Particle Size and Polydispersity Index

The use of a combination of lipids in the form of stearic acid and fat rich in monolaurin at various concentrations affected the particle size or the Z-average value. The combination of stearic acid and fat rich in monolaurin produced SLN with a small Z-average value; this was due to the ease of disintegration of larger particles and slower recrystallization in mixed lipids in the presence of fat rich in monolaurin due to its lower melting point and crystallinity index than stearic acid [31].

The results showed that the particle size of SLN increased with the increase in the ratio of stearic acid used; this was due to the stearic acid having a higher melting point (69.3 °C) than fat rich in monolaurin (43.2 °C). The higher melting point of stearic acid could lead to a less effective homogenization process, resulting in a larger size and wider particle size distribution [1,32]. Based on Table 3; it was known that the particle size of SLN in the formula of C3 has a smaller particle size compared to C1 and C2. The fat rich in monolaurin used in the C3 formulation was the highest concentration of 40% *w*/*w*; this showed the role of fat rich in monolaurin in homogenizing the emulsion system in SLN because it acted as an emulsifier, where increasing the concentration of fat rich in monolaurin could reduce the surface tension of the liquid lipid and water phases so that the mixture was more homogeneous and stable [33].

The particle size of the SLN produced at various concentrations of ferrous sulfate did not differ significantly; this was indicated by the particle size in the formulations C3, C4, and C5, which have sizes ranging from 278.70 to 318.10 nm; this was in line with research by Zariwala et al. [34], that the use of active ingredients at various concentrations in SLN preparation did not produce a significant change in SLN particle size. Perez et al. [19] also encapsulated hydrophilic compounds in the form of SLN using the double emulsion method and obtained SLN with a particle size of 277 to 550 nm.

The SLN particles produced in this study ranged from 278.70 to 540.40 nm. The size was in accordance with the range on the SLN, which ranges from 10–1000 nm; however, the size varies, which was indicated by a high polydispersity index (PI). If the PI value was close to 1.0, this indicated that the various particle sizes in the sample tested were less homogeneous and not monodisperse [35]. The preparation of SLN using a W/O/W emulsion system could cause the formation of several layers in the SLN particles, which can increase the size and variation, commonly named ’lipospheres’ [10]. In addition, the formation of several large SLNs can be caused by physical instability in the emulsion, which causes an increase in particle size during storage [36]. The high PI value obtained can also be caused by agglomeration. Fat rich in monolaurin, which was used as an emulsifier, is a non-ionic emulsifier that produced particles that were not charged. If the particle has no charge, hydrophobic fat interacts through attractive forces between other fat particles; this causes the formation of agglomerations and produces larger particle sizes.

#### 3.2.2. Entrapment Efficiency and Loading Capacity

Table 3 shows that the formulations C1, C2, and C3 with varying concentrations of fat rich in monolaurin used have a high entrapment efficiency (EE), about 99.97–99.99%. The combination of solid lipids could increase the loading of hydrophilic drugs due to the provision of an imperfect crystal lattice in the structure of the lipid phase. The combination of stearic acid and fat rich in monolaurin increased the proportion of amorphous in the solid lipid matrix, thereby reducing the overall crystallinity of the particles. The incorporation of fat rich in monolaurin into SLN could improve the crystal matrix so as to produce sufficient space to accommodate the active compound, thereby increasing the encapsulation capacity of the drug [37]; this indicated that the lipid matrix consisting of stearic acid (C18:0) and fat rich in monolaurin (C12:0), with different chain lengths, formed imperfect crystals; this imperfect crystal formation had more space to accommodate more ferrous sulfate.

The use of fat rich in monolaurin increased up to 40% in the C3 formulation, resulting in a high EE value. The use of ferrous sulfate at various concentrations (C3, C4, C5) also had no significant effect on high EE values; this shows that the increase in ferrous sulfate concentration up to 15% (C5) did not indicate a saturated concentration of ferrous sulfate to be encapsulated in SLN because EE showed that about 99% of ferrous sulfate was successfully encapsulated; this high EE value indicated more ferrous sulfate levels were coated in the SLN during the encapsulation process, so it can be said that the encapsulation process carried out in this study worked optimally.

The SLN from formula C5 with the highest active ingredient of ferrous sulfate was then analyzed for loading capacity. Based on the analysis of the content of Fe encapsulated in SLN particles using an atomic absorption spectrometer (AAS), the Fe content was obtained at 617.31 ppm, so the loading capacity value was 0.06%. Loading capacity obtained <1% indicates that the amount of Fe that can be encapsulated in a certain amount of SLN after the lyophilization process was small; however, when compared with the entrapment efficiency, the Fe content that was successfully encapsulated was about 99.99%; this shows that in the Fe encapsulation process, the amount of lipid matrix used was very large compared to the amount of Fe coated. Thus, the lipid matrix was able to encapsulate almost all of the Fe, but the levels of Fe in the resulting SLN particles were low.

The loading capacity of the SLN system tended to be lower because of the low solubility capacity of liquid lipids, especially the hydrophobic active compounds. Loading capacity could be increased by controlling the influencing factors, such as the solubility of the active compound in lipids, the miscibility of the active compound with lipids, the physicochemical properties of the solid lipid matrix, and the polymorphic state of the lipid material [38]. The main thing to note was that the solubility of the active compound in lipids must be high, so emulsifiers such as mono- and diacylglycerol were needed or with the addition of the solvents [39].

### 3.3. Morphology and Structure of SLN

Selected SLN particles from formula C5 were analyzed using scanning electron microscopy (SEM) equipped with energy dispersive X-ray (EDX) and Transmission Electron Microscopy (TEM) to confirm particle size, shape, and arrangement. The morphology of the SLN can be seen in Figure 2, while the shape and structure of the SLN can be seen in Figure 3.

Figure 2A showed that SLN-ferrous sulfate was spherical with a particle size of <1000 nm. Some of the SLN particles were seen to be attached and agglomerated, where there were clumps of small SLN particles that merged; this was consistent with the PI value obtained >1.0 (Table 3), which indicates the level of heterogeneous particle dispersion. The agglomeration that occurs could be caused by the lyophilization method used. In the drying process of SLN dispersion using a freeze dryer, a water phase was sublimated so that the resulting SLN tends to form a system (SLN deposit) rather than one free SLN particle.

The SEM results were then analyzed using energy dispersive X-ray spectroscopy (EDX/EDS) to determine the chemical composition of the sample surface [40]. The EDX showed that in spectrum 3, SLN particles have the main components in the form of carbon (C) = 76.92%, oxygen (O) = 22.93%, and ferrous (Fe) = 0.15% (Figure 1c). The high carbon and oxygen elements were obtained from the lipid matrix used in the SLN. In contrast, the ferrous elements were obtained from ferrous sulfate, which was successfully encapsulated in the SLN. The small Fe content indicated that the Fe encapsulated in SLN was relatively small compared to its coating component, namely lipids.

The TEM analysis showed that the SLN size was <500 nm (Figure 3B). The size obtained in the TEM analysis is much smaller than in the PSA analysis. Particle measurement using the PSA could be biased in such cases, where a single particle appears to be a group of much smaller particles [41]. The particle structure of SLN is depicted in Figure 3. Figure 3C shows a contrasting bright light pattern on the particles’ outer layer, indicating a surfactant layer around the nanoparticles. The black spots inside the particles in Figure 3A indicate Fe trapped in the lipid matrix [33]. Black spots scattered in the lipid matrix indicate that the Fe trapped in the SLN is evenly dispersed in the lipid matrix to form the nanospheres system in W/O/W emulsions [42]; these results can confirm that although the SLN polydispersity index is quite high (see Table 3), each SLN particle actually forms nanospheres that are smaller than the SLN particles.

Based on variations in the concentration of monolaurin-rich fat and ferrous sulfate used for the level of EE and the resulting physicochemical properties, it was known that the SLN-ferrous sulfate with the best results was shown by the C5 formulation, namely the use of 40% fat rich in monolaurin and the concentration ferrous sulfate was about 15%, which indicated by a small Z-average value of the particle, which was 292.40 nm, and a high EE value.

SLN-ferrous sulfate based on fat rich in monolaurin and stearic acid can be an alternative solution for iron fortification in food, where currently, direct iron fortification in the form of free ferrous sulfate is still constrained by instability and undesirable aftertaste; however, this SLN fabrication method still needs to be further developed, especially in improving the particle stability and increasing the loading capacity of SLN. 

## 4. Conclusions

Fabrication of SLN-ferrous sulfate in W/O/W emulsion system using fat rich in monolaurin and stearic acid as a solid lipid matrix was able to encapsulate ferrous sulfate well with high entrapment efficiency. The use of fat rich in monolaurin at a concentration of 40% *w*/*w* of lipids and ferrous sulfate at a concentration of 15% *w*/*w* obtained SLN with the best characteristics. SLN-ferrous sulfate had a Z-average value of 292.4 nm with a PI value of 1.03. SLN showed a spherical morphology, where the Fe trapped in the SLN was evenly dispersed in the lipid matrix to form a nanosphere system; however, this SLN fabrication method still needs to be further developed, especially in improving the particle stability and loading capacity.

## Figures and Tables

**Figure 1 nanomaterials-12-03054-f001:**
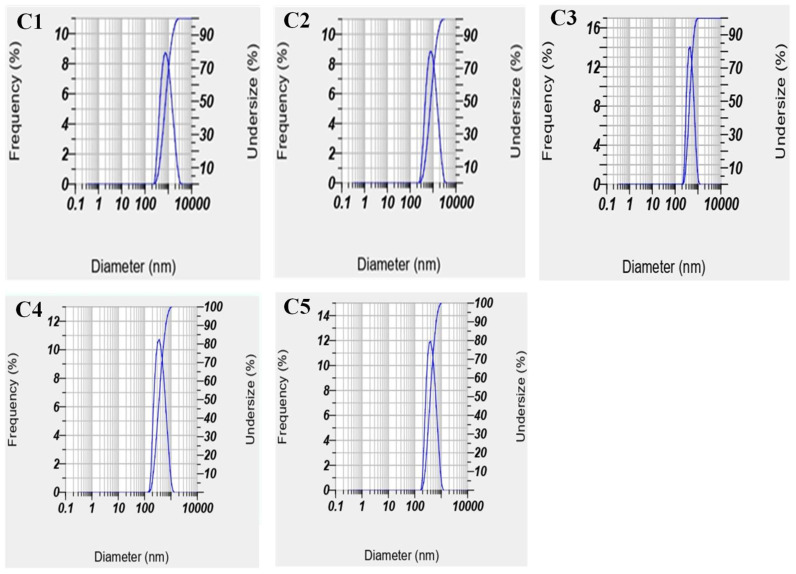
Particle size distribution of SLN-ferrous sulfate at various formulations (C1, C2, C3, C4, and C5) by particle size analyzer.

**Figure 2 nanomaterials-12-03054-f002:**
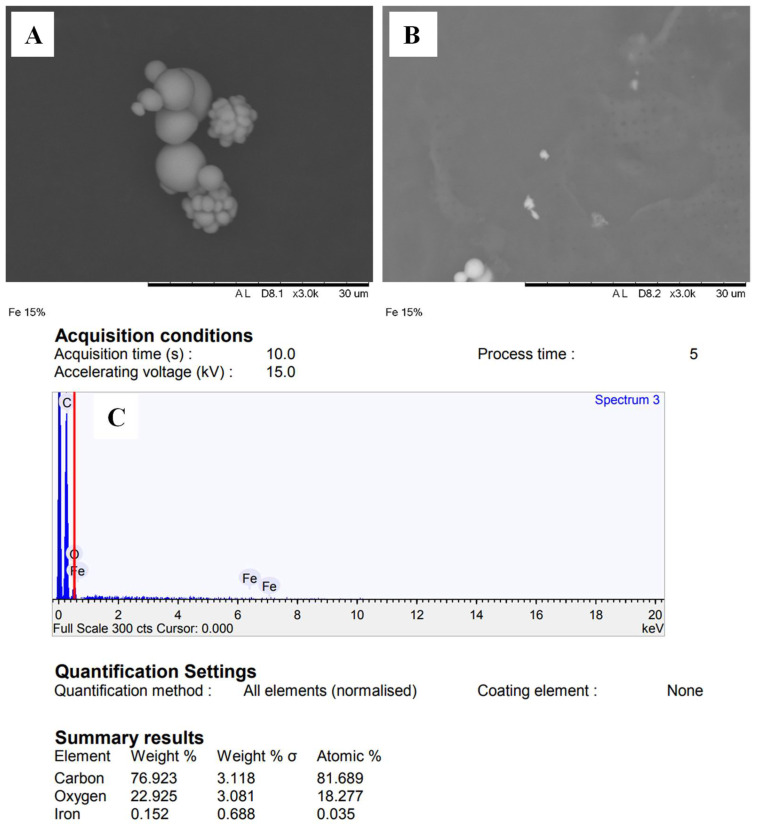
SLN morphology with 3000× magnification using SEM (**A**); SEM-EDX analysis area (**B**); and elements of SEM-EDX analysis (**C**).

**Figure 3 nanomaterials-12-03054-f003:**
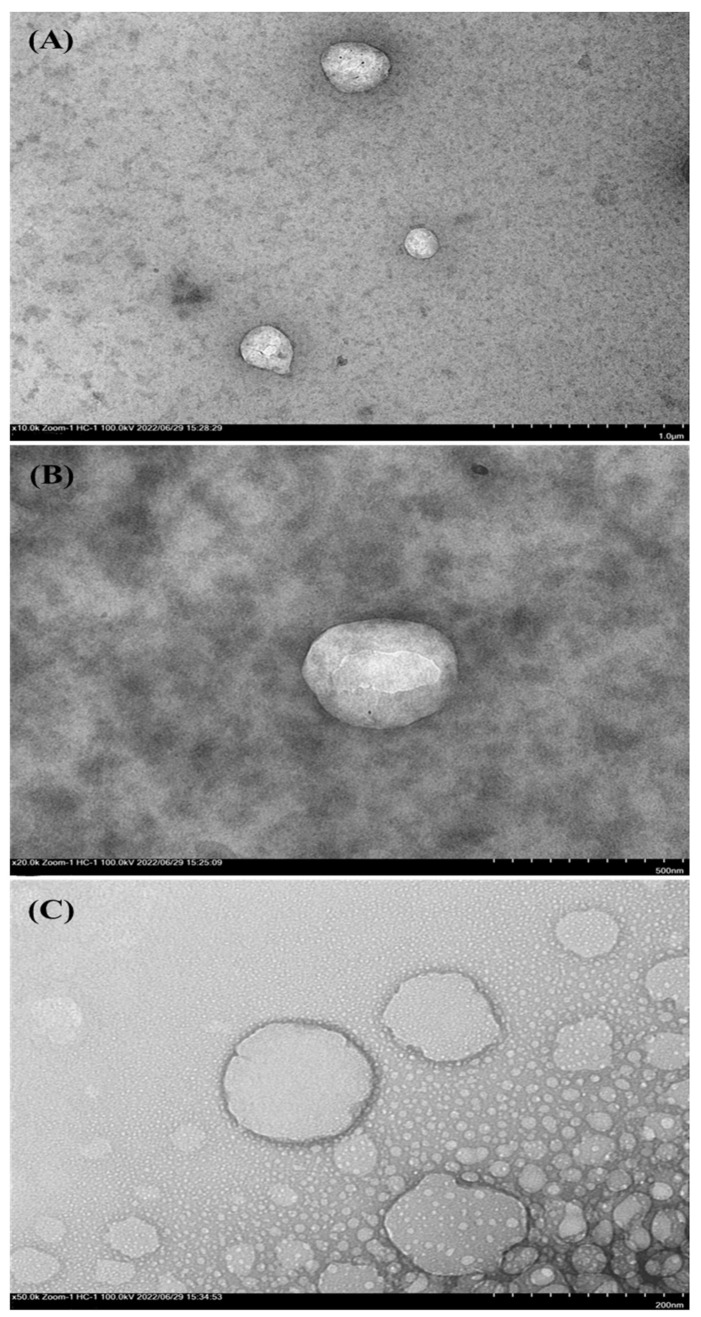
SLN structure with magnifications of 10,000× (**A**), 20,000× (**B**), and 50,000× (**C**) using transmission electron microscopy (TEM).

**Table 1 nanomaterials-12-03054-t001:** Formulation of SLN-ferrous sulfate.

Formula	Solid Lipid		Surfactant	Drug
	Fat Rich in Monolaurin (% *w*/*w*)	Stearic Acid (% *w*/*w*)	Tween80 (% *w*/*w*)	Ferrous Sulfate (% *w*/*w*)
C1	20	80	20	10
C2	30	70	20	10
C3	40	60	20	10
C4	40	60	20	5
C5	40	60	20	15

**Table 2 nanomaterials-12-03054-t002:** Characteristics of fat rich in monolaurin.

Characteristic	Value (%)
Composition of acylglycerol	
- Monoacylglycerol (MAG) - Diacylglycerol (DAG) - Triacylglycerol (TAG)	86.45 ± 11.48 13.55 ± 11.48 Not detected
Monolaurin content	40.52 ± 5.72
Emulsification capacity	95.26 ± 3.64

**Table 3 nanomaterials-12-03054-t003:** Z-average, polydispersity index, and entrapment efficiency of SLN-ferrous sulfate.

Formula	*Z*-Average (nm)	Polydispersity Index (PI)	Entrapment Efficiency (%)
C1	443.70	0.8830	99.99
C2	540.40	1.0450	99.97
C3	318.10	1.2440	99.99
C4	278.70	0.8910	99.99
C5	292.40	1.0300	99.99

## Data Availability

Not applicable.

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
