# Peer review of "Preparation of Solid Lipid Nanoparticle-Ferrous Sulfate by Double Emulsion Method Based on Fat Rich in Monolaurin and Stearic Acid"

_nanomaterials, 2022, doi:10.3390/nano12173054_

Round 1

Reviewer 1 Report

In this manuscript, the authors fabricated SLN-ferrous sulfate using a mixture of stearic acid and monolaurin-rich fat as a lipid matrix as well as an emulsifier using the double emulsion (W/O/W) method by varying the concentration of monolaurin-rich fat and ferrous sulfate concentration. Several parameters were studied such as particle size, entrapment efficiency, particle morphology, and microstructure of SLN. Result shows that SLN showed a spherical morphology, where the Fe trapped in the SLN was evenly dispersed in the lipid matrix to form a nanosphere system. Overall, it is helpful for researchers who are interested in this field. However, it is potentially publishable once the authors have addressed the following questions:

1.     The introduction is poor. The latest progress and background of solid lipid nanoparticles (SLN) are ambiguous. And some more important articles need to be cited, such as (Aaps Pharmscitech, 2011, 12(1): 62-76. Journal of drug targeting, 2012, 20(10): 813-830.).

2.     There are some problems with the format of the manuscript, for example the formulas in the manuscript on Page 3 do not have serial numbers. And the meaning of each parameter in the formulas needs to be explained.

3.     The scale bar of the TEM in Figure 3 is blurred.

4.     Figure 1 has some flaws. The starting position of the abscissa is too small to see the position of the peaks. And more explanation is needed to Figure 1 and Table 3.

Author Response

Response to Reviewer 1 Comments

In this manuscript, the authors fabricated SLN-ferrous sulfate using a mixture of stearic acid and monolaurin-rich fat as a lipid matrix as well as an emulsifier using the double emulsion (W/O/W) method by varying the concentration of monolaurin-rich fat and ferrous sulfate concentration. Several parameters were studied such as particle size, entrapment efficiency, particle morphology, and microstructure of SLN. Result shows that SLN showed a spherical morphology, where the Fe trapped in the SLN was evenly dispersed in the lipid matrix to form a nanosphere system. Overall, it is helpful for researchers who are interested in this field. However, it is potentially publishable once the authors have addressed the following questions:

Point 1: The introduction is poor. The latest progress and background of solid lipid nanoparticles (SLN) are ambiguous. And some more important articles need to be cited, such as (Aaps Pharmscitech, 2011, 12(1): 62-76. Journal of drug targeting, 2012, 20(10): 813-830.).

Response 1:

The latest progress and background of solid lipid nanoparticles (SLN) have been revised, and some more important articles such as (Aaps Pharmscitech, 2011, 12(1): 62-76, and Journal of drug targeting, 2012, 20(10): 813-830.) have been cited (Page 1-2, Line 40-60, and Page 10, line 350-351 and 354-355, red color).

Point 2: There are some problems with the format of the manuscript, for example the formulas in the manuscript on Page 3 do not have serial numbers. And the meaning of each parameter in the formulas needs to be explained.

Response 2:

The formulas in the manuscript on Page 3 have been added the serial numbers, and the meaning of each parameter in the formulas has been explained (Page 3-4, Line 122-134, red color).

Point 3: The scale bar of the TEM in Figure 3 is blurred.

Response 3:

Figure 3 has been replaced with a clearer image, and the scale bar of the TEM in Figure 3 is no longer blurred (Page 9).

Point 4: Figure 1 has some flaws. The starting position of the abscissa is too small to see the position of the peaks. And more explanation is needed to Figure 1 and Table 3.

Response 4:

Figure 1 is used as an illustration to help clarify the particle size distribution in Table 3, but the starting position of the abscissa cannot be changed because the figure is a default from a particle size analyzer (PSA). More explanations for Figure 1 and Table 3 have been added for more in-depth discussion. (Page 6, Line 203-220, red color)

Reviewer 2 Report

The synthesis and characterization of solid lipid nanoparticles containing ferrous sulfate and based on stearic acid and fat rich in monolaurin is described in this work. The results are presented and discussed in appropriate way and deserve to be published.

Author Response

Response to Reviewer 2 Comments

The synthesis and characterization of solid lipid nanoparticles containing ferrous sulfate and based on stearic acid and fat rich in monolaurin is described in this work. The results are presented and discussed in appropriate way and deserve to be published.

Response:

Thank You

Author Response

Response to Reviewer 3 Comments

This manuscript reports on encapsulating ferrous sulfate in solid lip nanoparticle based on stearic acid and fat-rich monolaurin by a double emulsion W/O/W method. This is a solid paper of good originality, technical quality and contribution to the field. The work is interesting, well-planned and correctly carried out. However, there are a few works needed overall on building the motivation in the introduction and application in the conclusion. There are also some concerns with the presentation of data in the figures, and the explanation of data is not thorough.

Point 1: The authors investigated entrapment efficiency and loading capacity of ferrous sulfate in SLN at various concentrations of monolaurin-rich fat (20, 30 and 40 % w/w lipid) and various concentrations of ferrous sulfate (5, 10, 15 % w/w lipid). Highest loading efficiency happened at the highest concentrations of both monolaurin-rich fat (40%) and ferrous sulfate (15%). Would the loading efficiency be improved with increasing the concentrations of monolaurin-rich fat and ferrous sulfate?

Response 1:

. Loading efficiency or loading capacity may be improved by increasing the concentrations of monolaurin-rich fat and ferrous sulfate, but the use of too much monolaurin-rich fat causes the lyophilization process to be longer and difficult because monolaurin-rich fat has the ability to bind water. While the use of higher ferrous sulfate may increase the loading efficiency, but it may have an impact on the amount of free iron that is not encapsulated or can reduce entrapment efficiency. Therefore we limit the use of monolaurin-rich fat and ferrous sulfate to concentrations of 40% and 15%, respectively.

Point 2: Highest loading capacity of ferrous sulfate was tested to be 0.06%. Is this value critically high enough to be stated as ‘a high efficiency’?

Response 2:

. In this study, the entrapment efficiency is already high, but the loading capacity is still quite low. However, the loading capacity may be increased by improving the fabrication method used, for example through the formation of nanostructured lipids that will form cavities in each nanoparticle to trap more iron, but this requires a more difficult technology. So the results of this study were good on entrapment efficiency, particle size, and particle morphology, but further research is needed to improve loading capacity.

Point 3: In the introduction part, the authors should compare the W/O/W method introduced in this study with other encapsulation approaches, such as electrospinning and electrospraying methods to emphasize reason of W/O/W method fits for the ferrous sulfate encapsulation. Some references should be cited to make the comparison:

Yao, Z.C., Chang, M.W., Ahmad, Z. and Li, J.S., 2016. Encapsulation of rose hip seed oil into fibrous zein films for ambient and on demand food preservation via coaxial electrospinning. Journal of food engineering, 191, pp.115-123. https://doi.org/10.1016/j.jfoodeng.2016.07.012

Yao, Z.C., Chen, S.C., Ahmad, Z., Huang, J., Chang, M.W. and Li, J.S., 2017. Essential oil bioactive fibrous membranes prepared via coaxial electrospinning. Journal of food science, 82(6), pp.1412-1422. https://doi.org/10.1111/1750-3841.13723

Yao, Z.C., Jin, L.J., Ahmad, Z., Huang, J., Chang, M.W. and Li, J.S., 2017. Ganoderma lucidum polysaccharide loaded sodium alginate micro-particles prepared via electrospraying in controlled deposition environments. International journal of pharmaceutics, 524(1-2), pp.148-158. https://doi.org/10.1016/j.ijpharm.2017.03.064.

Response 3:

In the introduction part, a comparison of the W/O/W method introduced in this study with other encapsulation approaches, such as electrospinning and electrospray methods for SLN-ferrous sulfate fabrication has been added, including:

Yao, Z.C., Chang, M.W., Ahmad, Z. and Li, J.S., 2016. Encapsulation of rose hip seed oil into fibrous zein films for ambient and on demand food preservation via coaxial electrospinning. Journal of food engineering, 191, pp.115-123. https://doi.org/10.1016/j.jfoodeng.2016.07.012

Yao, Z.C., Chen, S.C., Ahmad, Z., Huang, J., Chang, M.W. and Li, J.S., 2017. Essential oil bioactive fibrous membranes prepared via coaxial electrospinning. Journal of food science, 82(6), pp.1412-1422. https://doi.org/10.1111/1750-3841.13723

Yao, Z.C., Jin, L.J., Ahmad, Z., Huang, J., Chang, M.W. and Li, J.S., 2017. Ganoderma lucidum polysaccharide loaded sodium alginate micro-particles prepared via electrospraying in controlled deposition environments. International journal of pharmaceutics, 524(1-2), pp.148-158. https://doi.org/10.1016/j.ijpharm.2017.03.064.

(Page 2, Line 49-60, and Page 10-11, Line 356-363, red color)

Point 4: It should be ‘86.45 ± 11.48’ instead of ‘86.45 ± 11,48’ in Table 2., line 134, page 4.

Response 4:

. The writing of numbers on monoacylglycerol (MAG) content in Table 2 has been revised to '86.45 ± 11.48' (Page 4, Line 156-157, red color)

Reviewer 4 Report

A method has been developed for encapsulating iron sulfate in an emulsion matrix of fat rich in monolaurin and stearic acid. The best SLN performance was obtained using fat rich in monolaurin at a lipid concentration of 40% w/w. and ferrous sulfate at a concentration of 15% wt./mass. SLN showed a spherical morphology where Fe is uniformly dispersed in the lipid matrix.

The work is aimed at solving the important problem of fortifying food with iron and thus preventing anemia caused by iron deficiency. The results of the research are useful for making existing approaches and substantiating new directions.

The article can be recommended for publication, taking into account minor comments.

1.      The abbreviation used should be clarified in the text. For example, Z-average, TAG, MAG DAG and other similar.

2.      On fig. 1 shows the particle size distribution of SLN-ferrous sulfate at various formulations (C1, C2, C3, C4, and C5) by particle size analyzer. However, the logarithmic scale does not show the difference in particle sizes. The table on this page is quite sufficient and informative.

3.      It is desirable to indicate how the results of the study correlate with the solution of the specific problem of food iron fortification and the prospects for the development of the developed method.

Author Response

Response to Reviewer 4 Comments

A method has been developed for encapsulating iron sulfate in an emulsion matrix of fat rich in monolaurin and stearic acid. The best SLN performance was obtained using fat rich in monolaurin at a lipid concentration of 40% w/w. and ferrous sulfate at a concentration of 15% wt./mass. SLN showed a spherical morphology where Fe is uniformly dispersed in the lipid matrix.

The work is aimed at solving the important problem of fortifying food with iron and thus preventing anemia caused by iron deficiency. The results of the research are useful for making existing approaches and substantiating new directions.

The article can be recommended for publication, taking into account minor comments.

Point 1: The abbreviation used should be clarified in the text. For example, Z-average, TAG, MAG DAG and other similar.

Response 1:

. The abbreviations used have been clarified in the text.

Point 2: On fig. 1 shows the particle size distribution of SLN-ferrous sulfate at various formulations (C1, C2, C3, C4, and C5) by particle size analyzer. However, the logarithmic scale does not show the difference in particle sizes. The table on this page is quite sufficient and informative.

Response 2:

Yes, the logarithmic scale in Figure 1 does not clearly show the difference in particle sizes. Figure 1 only shows a general description of the particle size distribution of SLN-ferrous sulfate at various formulations. Therefore, it is confirmed with numerical data, as shown in Table 3, to make it more sufficient and informative.

Point 3: It is desirable to indicate how the results of the study correlate with the solution of the specific problem of food iron fortification and the prospects for the development of the developed method.

Response 3:

The correlation of the results of this study with the solution of problems related to iron fortification in food and the prospects for developing methods have been added to the Discussion section and Conclusions section. (Page 8, Line, 303-307, and Page 10, Line 319-321, red color)

Reviewer 5 Report

The authors tried to prepare solid lipid nanoparticle-ferrous sulfate by double emulsion method based on fat rich in monolaurin and stearic acid. They also stated that this SLN-ferrous sulfate by double emulsion method based on stearic acid and fat rich in monolaurin effectively encapsulated ferrous sulfate with high entrapment efficiency and good physicochemical properties. This study is interesting. However, there are many issues that need to be solved.

1. The value of the polydispersity index (PI) is too large. Normally, when this value is more than 0.5, there is meaningless. The authors need to take care of this problem.

2. The authors only utilized TEM and SEM and such methods to analyze the morphology and size properties, the characterization methods are too limited for the whole study.

3. The TEM and SEM results are not given positive results for the whole study, which can not let the audiences trust the results very well. It just looks like some random particles.

Author Response

Response to Reviewer 5 Comments

The authors tried to prepare solid lipid nanoparticle-ferrous sulfate by double emulsion method based on fat rich in monolaurin and stearic acid. They also stated that this SLN-ferrous sulfate by double emulsion method based on stearic acid and fat rich in monolaurin effectively encapsulated ferrous sulfate with high entrapment efficiency and good physicochemical properties. This study is interesting. However, there are many issues that need to be solved.

Point 1: The value of the polydispersity index (PI) is too large. Normally, when this value is more than 0.5, there is meaningless. The authors need to take care of this problem.

Response 1:

The value of the polydispersity index (PI) is indeed large (>0.5), which indicates that the resulting particle sizes are diverse (non-uniform). This was caused by the agglomeration of SLN particles, as described in the discussion. However, the resulting average particle size is quite small, which is indicated by the Z-average is relatively small (<300 nm). In addition, the resulting SLN particles also show the formation of nanospheres, as confirmed in Figure 3, so that in each SLN particle there are actually nanospheres that are smaller than the particles. (Page 6, Line 208-220, and Page 7, Line 290-292, red color)

Point 2: The authors only utilized TEM and SEM and such methods to analyze the morphology and size properties, the characterization methods are too limited for the whole study.

Response 2:

SLN morphology was analyzed by scanning electron microscopy equipped with energy dispersive X-ray spectroscopy (SEM-EDX) and transmission electron microscopy (TEM) to determine the characteristics, especially shape and size. This instrument is a sophisticated analytical instrument and has been able to confirm the results of other analyzes that have been carried out using a particle size analyzer (PSA) for particle size distribution and atomic absorption spectroscopy (AAS) for iron content in SLN.

Point 3: The TEM and SEM results are not given positive results for the whole study, which can not let the audiences trust the results very well. It just looks like some random particles.

Response 3:

The TEM and SEM results show that the morphology of the shape and size of the particles is small (<300 nm) but quite diverse. This also confirms the polydispersity index data, which is still quite high. This may be a suggestion for the development of further research to improve particle uniformity, as written at the end of the discussion (Page 8, Line 303-307, red color). However, the results of this study can be used as a reference for the development of SLN fabrication research by utilizing fat rich in monolaurin, which has a better functionality value than other saturated fats, which can be used to encapsulate the active compound ferrous sulfate or other active compounds.

Round 2

Reviewer 1 Report

Accept

Author Response

Reviewer 1 Comments: Accept

Response: Thank You

Reviewer 5 Report

The manuscript is OK now.

Author Response

Reviewer 5 Comments: The manuscript is OK now

Response: Thank You